# Number-resolved imaging of $^{88}$Sr atoms in a long working distance optical tweezer

Niamh Christina Jackson[1], Ryan Keith Hanley[1,2], Matthew Hill[1],
Frédéric Leroux[1], Charles S. Adams[1] and Matthew Philip Austin Jones[1⋆]

**1** Joint Quantum Centre (Durham-Newcastle), Department of Physics,
Durham University, DH1 3LE, United Kingdom
**2** Department of Physics, University of Oxford, Clarendon Laboratory,
Parks Road, Oxford, OX1 3PU, United Kingdom

⋆ m.p.a.jones@durham.ac.uk

## Abstract

We demonstrate number-resolved detection of individual strontium atoms in a long working distance low numerical aperture (NA = 0.26) tweezer. Using a camera based on single-photon counting technology, we determine the presence of an atom in the tweezer with a fidelity of 0.989(6) (and loss of 0.13(5)) within a 200 µs imaging time. Adding continuous narrow-line Sisyphus cooling yields similar fidelity, at the expense of much longer imaging times (30 ms). Under these conditions we determine whether the tweezer contains zero, one or two atoms, with a fidelity > 0.8 in all cases, with the high readout speed of the camera enabling real-time monitoring of the number of trapped atoms. Lastly we show that the fidelity can be further improved by using a pulsed cooling/imaging scheme that reduces the effect of camera dark noise.

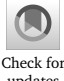

# 1 Introduction

Methods for isolating and reading out individual quantum systems are at the heart of current developments in quantum science and technology. Individually trapped neutral atoms were first observed in a magneto-optical trap (MOT) [1], and then in optical tweezers [2, 3] and optical lattices [4]. Since then, the optical tweezer approach has been developed to produce addressable arrays of arbitrary geometry [5] and dimensionality [6–9] containing $N \approx 100$ atoms. Applications include quantum simulation [10, 11] and computation [12], as well as quantum chemistry [13, 14].

A key recent development was the extension of tweezer array techniques from alkali-metal atoms to the divalent atomic species Sr [15, 16] and Yb [17]. These species have important applications in optical frequency standards [18] due to their extremely narrow (< 1 Hz) optical clock transitions. In combination with tweezer array technology, this highly coherent environment [19, 20] offers new perspectives in quantum-enhanced metrology and quantum simulation. Furthermore, narrow intercombination cooling transitions provide powerful new methods for loading [15, 16] and high-fidelity imaging in tweezer arrays [21], as well as cooling to the motional ground state [15, 16, 22]. In all tweezer array experiments so far, the tightly-focused tweezers were created with high-numerical aperture (NA > 0.5) lenses with working distances of < 15 mm. This inevitably leads to the presence of dielectric surfaces close to the trapped atoms, with the potential for unwanted systematic shifts of the ultra-narrow clock transitions [23, 24].

In this paper we present the isolation and detection of individual strontium atoms in an optical tweezer with a working distance of 37 mm (NA = 0.26). Combined with a conductive coating on the lenses and in-vacuo electrodes, this system is designed to provide a tweezer array platform compatible with precision measurement, and in particular with our proposal to create non-classical states in optical atomic clocks using Rydberg states [25]. We observe that it is possible to load ultra-cold atoms into the tweezer directly from a magneto-optical trap operating on a narrow intercombination line, even when the differential AC Stark shift on the cooling transition is significant, in agreement with the results in [21].

For imaging, we investigate a newly available type of camera based on an array of single-photon-counting avalanche photodiodes. The low readout noise and high frame rate enables the presence of an atom to be determined with 0.989(6) fidelity in an exposure time of just 200 µs. By employing existing Sisyphus cooling techniques [15, 21], we show that it is possible to identify two atoms in the tweezer with a fidelity of 0.83(2), but at the expense of a longer imaging time (30 ms) and higher loss. In this regime, we demonstrate real-time monitoring of the number of trapped atoms, enabling one-body loss due to dark state pumping to be observed as discrete jumps in the atom number. In future experiments, the ability to measure the number of atoms as a function of time could provide a route to quasi-deterministic preparation of single-atom states without transport or light-assisted collisions [26–28]. Lastly, we demonstrate further improvements to the fidelity of single atom detection by using an alternating cooling-imaging sequence to eliminate the effect of camera dark noise.

# 2 Long-working distance optical tweezer

A schematic of the experiment is shown in Fig. 1, and a detailed description is provided in [29]. The optical tweezer is created by focusing a 532 nm trapping beam to a waist of $w_r = 1.28(1)$ µm using a custom aspheric lens with a numerical aperture of 0.26, mounted inside an ultra-high vacuum (UHV) chamber. The corresponding Rayleigh length is $z_0 = 9.68(15)$ µm, and typical radial and axial trap frequencies for a trap depth of 1 mK are

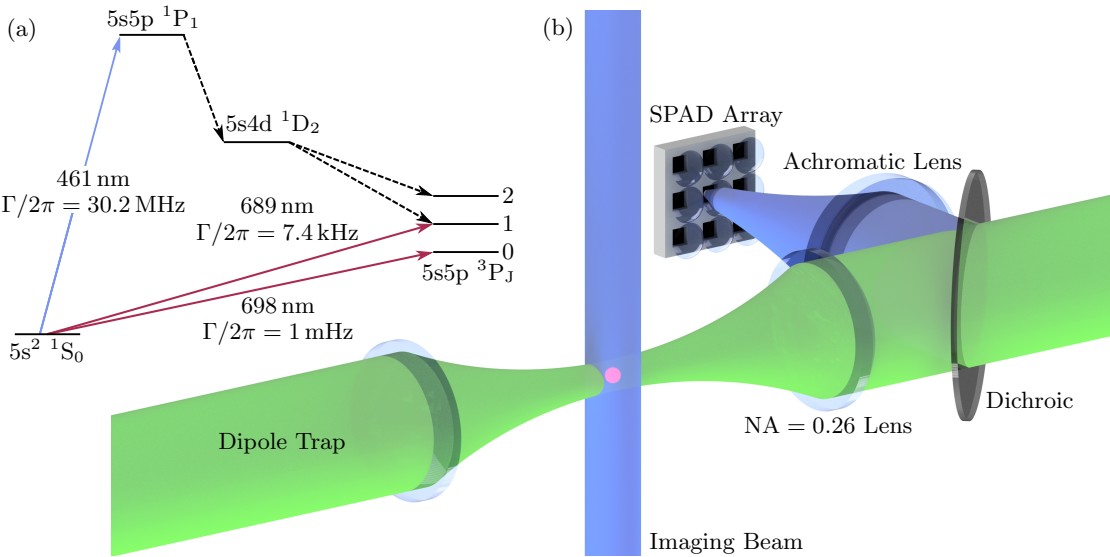

Figure 1: (a) Sr energy level diagram. (b) Experimental set-up. Strontium atom(s) are trapped in a 532 nm optical tweezer formed by the NA = 0.26 aspheric lenses. A 461 nm probe beam is used to image the atom(s), with the resulting fluorescence collected through one of the aspheric lenses and imaged onto the SPAD camera via its integrated microlens array.

$\omega_{\mathrm{r}} = 2\pi \times 76$ kHz and $\omega_{\mathrm{a}} = 2\pi \times 7$ kHz respectively. The lens has a broadband anti-reflection coating on the input side, and a transparent conductive indium tin oxide (ITO) coating on the side facing the atoms. Such ITO coatings have proven important in reducing stray electric fields in previous experiments with Rydberg atoms in optical tweezers [30]. The lens design was optimised for trapping at 532 nm and at the 813 nm magic wavelength for the clock transition, as well as for collection of the fluorescence at 461 nm. On the opposite side of the UHV chamber, an identical objective collects and recollimates the trapping light. Between the lenses, two planar arrays of 6 electrodes enable electric fields to be applied along all the available optical axes. A pair of coils mounted inside the vacuum provide a strong quadrupole field for the MOT, as well as the ability to apply static fields of up to 8 mT. The latter will enable us to exploit the clock transition in the bosonic isotopes of strontium [31–33].

Compared to similar optical tweezer setups based on in-vacuo lenses, the working distance in our experiment ($d = 37$ mm) is a factor of $> 2$ larger [7]. The difference is even more significant compared to experiments based on air-side objectives and glass cells [10,15,16], where the atom-surface distance may be only a few millimetres. This feature of our apparatus is important since unwanted DC Stark shifts due to the presence of nearby surfaces have led to significant problems in Rydberg experiments [34,35] and optical atomic clocks [23]. For example, recent experiments on Rydberg quantum simulation [10] required near-continuous application of UV light to remove Rb atoms from the cell surface in order to control the electric field. However this Light-Induced Atomic Desorption (LIAD) [36] technique has not been demonstrated for atoms like Sr and Yb. Even for conductive surfaces, the presence of adsorbates may lead to significant unwanted fields [37]. Work function differences between adjacent materials may also lead to stray fields. The choice of a longer working distance was thus motivated by the potential for a significant reduction in stray field, since the DC Stark shift due to small patch potentials will scale as $d^{-4}$. As we show in section 6, we are still able to detect single atoms with high fidelity despite the reduced collection efficiency compared to high-NA designs. As a side benefit, the use of lower NA lenses to trap single atoms also provides improved optical access for cooling and probe beams.

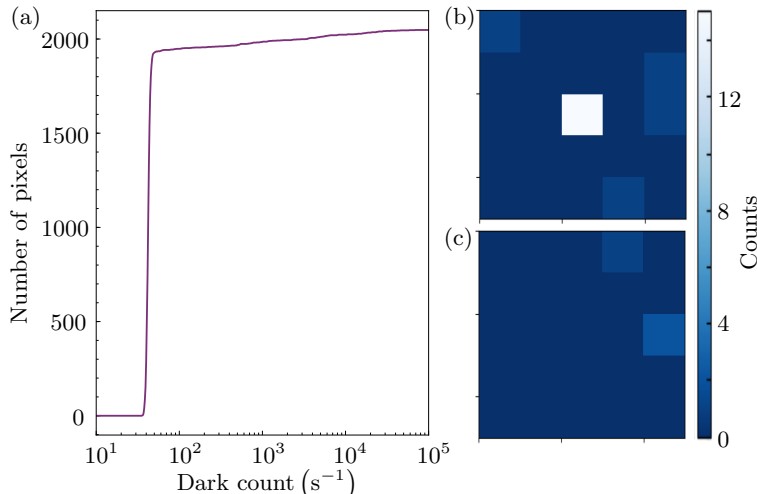

Figure 2: (a) Measured cumulative distribution function of the dark count over the SPAD array. (b) and (c) Images obtained on the sensor after 1 ms of exposure time, over an area of 5 × 5 pixels, in the presence (b) and absence (c) of a single atom.

## 3  Avalanche Photodiode Array Detector

Conventionally, tweezer array experiments rely on either single-pixel Single Photon Avalanche Diode detectors (SPADs), or on intensified or electron-multiplying CCD cameras as detectors. Recently sCMOS cameras have also been investigated [38]. Single-pixel SPADs have the advantage of true photon counting and nanosecond time resolution, making them useful for experiments in quantum optics, but are difficult to scale to large numbers of traps. Conversely, CCD or sCMOS cameras have higher noise and relatively slow frame rates, but enable the simultaneous readout of arrays containing thousands of traps if required.

In this paper we follow a different approach, based on a commercially available SPAD array detector (Micro Photon Devices SPC3). The array consists of 64 × 32 pixels, each of which is an independent SPAD [39]. Independent counters are provided for each pixel that return the number of detected photons within the gated exposure time, which may be as short as 1.5 ns. Full-frame readout of the camera is possible at $9.6 \times 10^4$ frames per second. The measured dark count distribution is shown in Fig. 2(a), with > 95% of the pixels having a dark count below 100 s$^{-1}$. The measured quantum efficiency at 461 nm is 36(2)% and the signal-to-noise ratio is 1.9 at 461 nm (for 10 incident photons per pixel). Compared to CCD or sCMOS technology (see [38] for a useful comparison), the SPAD array detector has the edge for short exposures and low numbers of detected photons, where EMCCD and sCMOS cameras are limited by readout noise. As the exposure time (and number of photons) increases, the higher quantum efficiency and lower dark count rate of EMCCD and sCMOS cameras ultimately wins out. However, the SPAD camera still retains the advantages of fast gating and readout.

An important difference between the SPAD array and an EMCCD camera is the pixel size. The SPAD pixels are large (150 μm × 150 μm), with an active area of 30 μm × 30 μm at the centre. An integrated microlens in front of each pixel boosts the collection efficiency to 85% of the chip surface. Nevertheless, the large pixel size means a substantial overall magnification is required. For the experiments presented here with a single tweezer, we use an overall magnification of ten, giving an effective pixel size in the object plane of 15 μm. Thus all the light from the dipole trap is concentrated on a single pixel. For future experiments with trap arrays, the magnification will be boosted by an additional telescope. In tandem with control over the array spacing using spatial light modulators, the SPAD array should provide a flexible

readout device for arrays of > 1000 traps.

## 4   Loading the tweezer

The optical tweezer is loaded from a narrow-line magneto optical trap (nMOT) operating on the $5s^2\,^1S_0 \rightarrow 5s5p\,^3P_1$ line at 689 nm. The nMOT itself is loaded using the conventional sequence of pre-cooling on the broad $5s^2\,^1S_0 \rightarrow 5s5p\,^1P_1$ transition at 461 nm, followed by transfer to a "broadband" cooling phase of duration 150 ms on the 689 nm transition. The tweezer beam is turned on at a time $t_{\text{load}}$ before the end of the nMOT phase, which lasts for 100 ms. Varying $t_{\text{load}}$ enables relatively precise control over the mean atom number in the tweezer.

After a variable hold time, atoms in the tweezer are imaged using of light on the $5s^2\,^1S_0 \rightarrow 5s5p\,^1P_1$ transition, with the resulting fluorescence collected and imaged onto the SPAD array as shown in Fig. 2. For the data in this section and the next, where the trap was loaded with many atoms, the blue cooling beams were used as the imaging light. To get down to single atom sensitivity, we added an additional probe beam as shown in Fig. 1 which propagated orthogonal to the trap beam. Empirically we find that this alignment with the axis of tightest confinement is essential to avoid atoms being pushed along the trap axis during imaging.

To our surprise, we found that direct loading from the nMOT was possible even for deep tweezers where the differential AC Stark shift on the cooling transition far exceeded the linewidth. Efficient loading also occurs despite the relatively small number of photons that can be scattered during the time it takes a 1 μK atom in the nMOT to cross the tweezer. We attribute this efficient loading to the presence of a substantial fraction of atoms in the tail of the Boltzmann distribution that are moving slowly enough to scatter many photons [29]. We find that for trap depths $U_0/k_{\text{B}} > 30$ μK, atoms can also be loaded into subsidiary intensity maxima formed by diffraction of the trapping beam by the circular aperture of the aspheric lenses [29]. To avoid these effects, we first loaded atoms into a tweezer of depth $U_0/k_{\text{B}} = 10$ μK, before ramping to the final tweezer depth $U_{\text{F}}$ over a time 1 s.

To measure the temperature of the trapped atoms, we extended two methods previously developed for alkali-metal atoms. The first is the conventional ballistic expansion technique. In order to achieve sufficient signal-to-noise, these experiments are carried out with a large number of trapped atoms, though the method has been applied to single atoms [40]. The second method is a release and recapture technique described in [41]. Here, the trap is turned back on at some point during the expansion to recapture the atoms. A temperature is extracted from the measured decay of the recaptured atom number as a function of expansion time using comparison with a Monte-Carlo simulation. Note that both of these methods measure the temperature in the radial direction.

A typical release and recapture signal is shown in Fig. 3(a), along with the best fit from the Monte-Carlo simulation which yields a temperature of 24.0(1.0) μK in a 520 μK deep tweezer. This result is in excellent agreement with that obtained from the ballistic expansion method (24.8(4) μK) for the same trap. We note that this agreement is only found if the trap depth is ramped. For the case where atoms are loaded directly into deep traps, both methods yield unreliable results due to the presence of colder atoms trapped in subsidiary maxima, as shown in 3(b). Here the data taken without ramping show a distorted ballistic expansion, with very cold atoms trapped in the shallow subsidiary maxima dominating at long times. This effect is also visible in images of the ballistic expansion [29].

Lastly, we note that due to the very efficient cooling that occurs during loading, and the adiabatic nature of the ramp, we are able to prepare very cold atoms in deep traps. We find

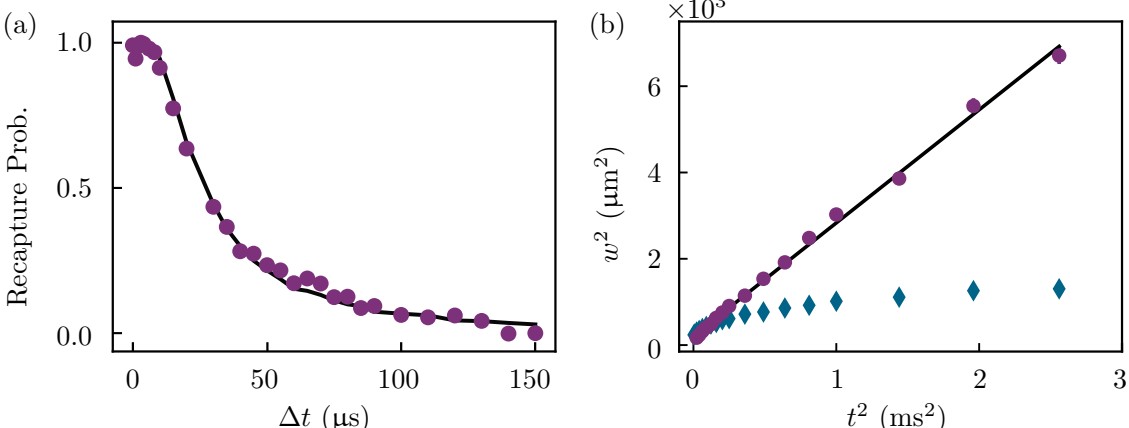

Figure 3: (a) Release and recapture measurement after ramping the trap depth to 520 μK (purple circles), compared to the fit from a Monte-Carlo simulation (blue line). (b) Ballistic expansion measurement in a ramped trap (purple circles) and without ramping (blue diamonds), for a final trap depth of 520 μK. The temperature obtained from the fit (black line) agrees with that obtained in (a).

that we can achieve a ratio $U_F/(k_B T) \approx 50$ for traps up to 5 mK deep.

## 5 Characterizing the tweezer

For cold atoms trapped in the harmonic part of the tweezer potential, the key parameters describing the trapping are the trap depth $U_F$ and the radial and axial trap frequencies $\omega_r$ and $\omega_z$. Independent measurements of these quantities yield information on the trapping beam such as the waist size.

In recent work with Sr atoms, trap frequencies were empirically obtained by observing motional sidebands on the $5s^2\,{}^1S_0 \rightarrow 5s5p\,{}^3P_1$ transition [16]. However this technique only works well in a magic-wavelength trap, where the upper and lower states have the same polarizability (and hence the same harmonic energy level spacing). In larger dipole traps parametric heating is often used [42], but we find in common with others that this method does not work so well for deep tweezer potentials with high axial confinement. Instead we generalize a release-and-recapture technique developed for alkali-metal atoms [43].

The technique involves turning the trap off for two short periods of fixed duration $t_1$ and $t_2$, separated by a variable duration $\Delta t$ where the trap is on. The first dark period $t_1$ imparts a well-defined phase to the oscillations in the trap; the subsequent probability of losing the atoms during $t_2$ depends on whether the atoms are at a turning point of their motion and hence on the trap frequency.

Examples of the resulting oscillations in the recapture probability are shown in the insets in Fig. 4. Since the atoms are significantly colder than in previous work with Rb, the optimal release times were found to be longer with $t_1 = 5\,\mu s$ (25 μs) and $t_2 = 20\,\mu s$ (60 μs) for a trap depth of 1.2 mK (60 μK). The data in Fig. 4 are well described by a damped sine wave, from which we obtain a measurement of the radial trap frequency after correcting for the damping.

Fig. 4 shows $\omega_r^2$ as a function of the trap power $P$ over a large range of trap depths. From the gradient it is possible to extract the trap waist $w_0$. In the harmonic approximation, the trap frequency as a function of power is given by $\omega_r^2 = 4\alpha_0 P/(m\pi\epsilon_0 c w_0^4)$ where $\alpha_0$ is the ground state polarizability. The value of $\alpha_0$ is dominated by the strong $5s^2\,{}^1S_0 \rightarrow 5s5p\,{}^1P_1$ transition, and can be calculated to high accuracy. From the gradient of the fit in Fig. 4 we find

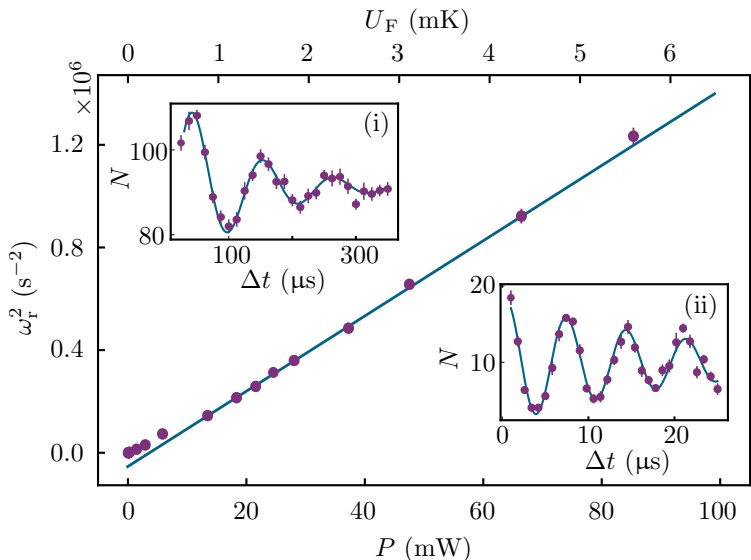

Figure 4: The square of the radial trap frequency $\omega_r^2$ as a function of the trap power $P$ (lower axis) or trap depth $U_F$ (upper axis). The fit to the data is a straight line (dark blue) which excludes the data at low trap powers. Insets show the trap frequency measurements taken at two different powers of (i) 0.3 mW (60 μK) and (ii) 20 mW (1.2 mK) after ramping, where $N$ is the number of trapped atoms.

$w_0 = 1.28(1)$ μm. We exclude points at lower trap powers, as otherwise we obtain a poor fit to the data. We attribute the poor fit at lower trap powers to the small ratio of the atomic temperature to trap depth, rendering the harmonic approximation invalid. This is further highlighted by the insets, where significantly higher damping is seen at lower trap powers.

## 6 Number-resolved imaging

The performance of our imaging system in detecting single atoms was characterised by comparing different imaging techniques. In each case the atom(s) are prepared in a tweezer of depth $U_F/k_B = 7.5$ mK. Unless otherwise stated, the probe beam shown in Fig. 1 is tuned to the AC Stark-shifted resonance of the trapped atom(s). The frequency shift is determined spectroscopically, with a measured shift of 26(1) (56(1)) MHz/mK for the $|m_j| = 1$ ($|m_j| = 0$) states. The vertically propagating imaging beam is linearly polarised in the horizontal direction, for maximal coupling to the $|m_j| = 1$ state. The atom number in the tweezer is controlled by varying $t_{load}$. Typically, imaging a single atom requires $t_{load} < 5$ ms.

**Imaging with continuous Sisyphus cooling**

The first method presented in Fig. 5 follows the approach developed in [15]. Atoms are imaged using light on the 461 nm transition while being simultaneously laser cooled on the 689 nm transition via an in-trap Sisyphus cooling mechanism. A similar technique using sideband cooling is described in [16]. In our experiment, Sisyphus cooling was achieved by continuously applying the 689 nm MOT beams during imaging. To balance the low cooling rate on this narrow transition, the scattering rate on the 461 nm transition was lowered by reducing the saturation parameter to $S = 0.004$, and by detuning the probe beam by -20 MHz from the AC Stark shifted resonance. At our trap wavelength of 532 nm, the 5s5p $^3P_1$ state experiences a

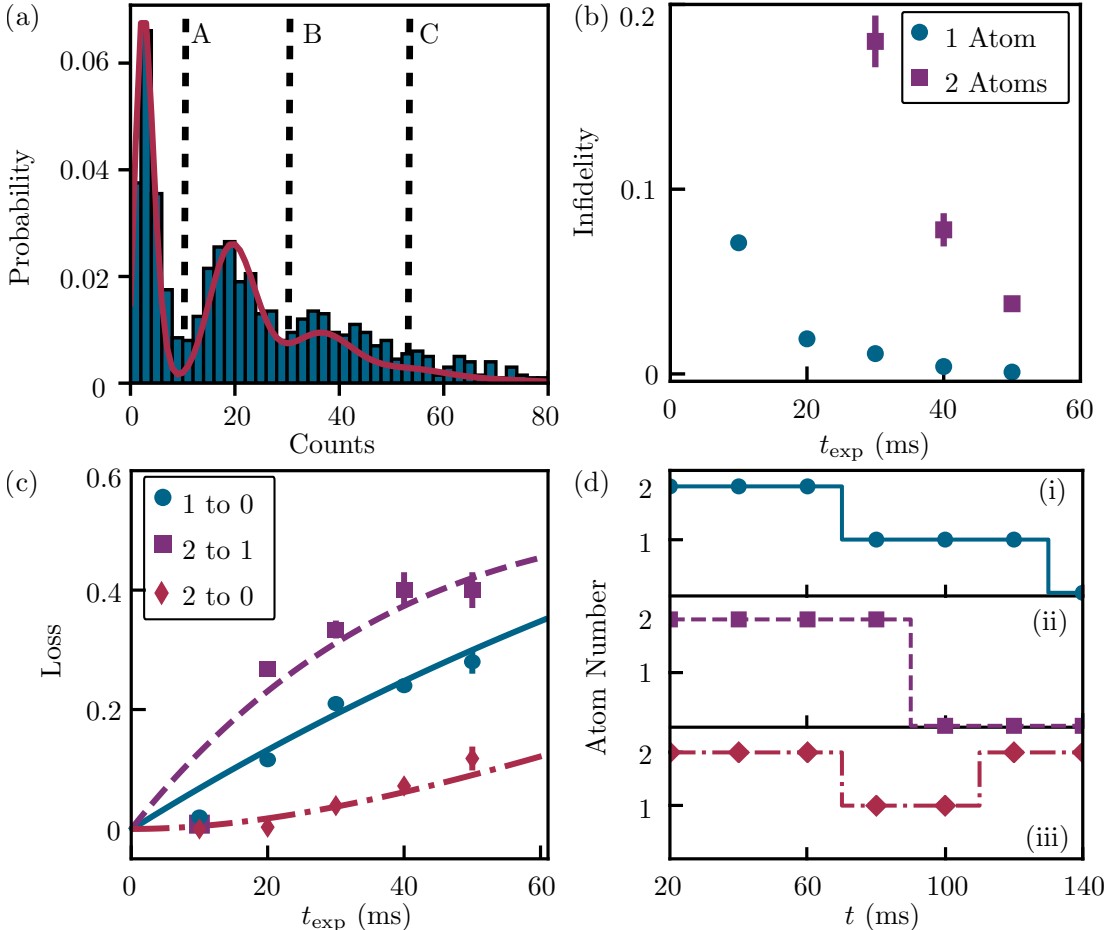

Figure 5: (a) Histogram of the number of counts for $t_{exp} = 30$ ms and 1000 experimental repetitions. The bin width is 2 counts. The red solid line shows the composite Poisson model. Vertical dashed black lines indicate the optimised thresholds used to calculate the infidelities and loss rate. (b) Limiting infidelity for a single atom (blue circles) and selective infidelity for two atoms (purple squares) as a function of $t_{exp}$. (c) Loss versus $t_{exp}$ for $1 \rightarrow 0$ (blue circles), $2 \rightarrow 1$ (purple squares) and $2 \rightarrow 0$ (red diamonds). Blue solid, purple dashed and red dash-dotted lines indicate the respective predicted losses due to depumping into the $^1D_2$ state. (d) Atom trajectories showing specific cases of decay from (i) two to one to zero (ii) two to zero atoms and (iii) where an error occurs

smaller AC Stark shift than the ground state [29]; Sisyphus cooling thus occurs via the "repulsive" mechanism identified in [15]. For our parameters, the optimum detuning for the 689 nm beams was determined to be $\Delta_{689}/(2\pi) = +5.9$ MHz. For each realisation of the experiment, fifteen consecutive 10 ms frames were recorded. The inter-frame delay was < 10 ns. The frames can be analysed separately, or combined to form a cumulative exposure of duration $t_{exp}$.

The histogram in Fig. 5(a) shows the number of counts obtained for a cumulative imaging time $t_{exp} = 30$ ms formed from the first three frames. Three peaks are clearly visible, corresponding to $N = 0, 1, 2$ atoms in the tweezer. In previous work with Sr [15,16], light-assisted collisions were deliberately introduced in order to prepare traps containing either zero or one atom. In [15] this was achieved using near resonant light on the 689 nm transition. However, in [16], a total of 2 ms of light on the 461 nm transition was used to achieve the same ef-

fect. Since we image for a longer times, and with a higher intensity, it might be expected that light-assisted collisions would also occur in our experiment. The major difference between our experiment and [16] is that the volume of our tweezer $V \propto w_r^2 z_0$ is 15 times larger. Since the light assisted collision rate goes as the square of the density, this leads to a negligible collision rate during our imaging duration.

The fit to the histogram in Fig. 5(a) is based on a Poisson distribution for both the trapped atom number and the number of photons detected per atom in each experimental run. The resulting computer-generated distributions are corrected for one-body loss due to the weak decay channel $5s5p\,^1P_1 \rightarrow 5s4d\,^1D_2$ [44]. At 532 nm the 5s4d $^1D_2$ state is strongly anti-trapped, therefore any decay into this state leads to loss. To include this decay, a weighted average of histograms is performed, with the weighting reflecting the exponential decay of the atom number during the imaging time. The mean atom number $\bar{N} = 1.2$ is obtained from the experimental data, while the effective decay rate is calculated using the branching ratio [15, 16] along with the scattering rate of the imaging beam. Therefore, the only free parameter is the mean number of detected photons per atom, $\alpha = 17$. The model is in reasonable agreement with the experimental data, supporting a Poissonian description of the loading statistics.

To quantify the performance of our imaging system, we have extended the method developed in [15,16,21] based on the analysis of atom number correlations between two successive imaging frames. The atom number in each frame was determined by applying three thresholds (labelled A, B and C in Fig. 5(a)), such that $n_c < A = 0$, $A < n_c < B = 1$, $B < n_c < C = 2$, where $n_c$ is the number of counts, as shown in 5(a).

In previous work, a parity projection step restricted the atom number to either zero or one [15, 16, 21]. Two types of error were considered: an infidelity error occurs when atoms are observed in the second image, but not in the first, and a loss error occurs when atoms are present in the first image but not in the second. We performed a similar analysis by excluding events with $N > 1$ using threshold (B). In this limit, the fidelity $F_1$ and loss $L_1$ are given by

$$F_1 = 1 - \frac{P(01)}{P(0) + P(1)} \tag{1}$$

$$L_1 = \frac{P(10) - P(01)}{P(1)}, \tag{2}$$

where $P(m)$ is the probability of loading $m$ atoms and $P(mn)$ is the probability of identifying $m$ atoms in the first frame and $n$ atoms in the second. This limiting fidelity $F_1$ is useful for comparison between experiments, since it is independent of the mean atom number.

To evaluate the accuracy with which the atom number ($N = 0$, 1 or 2) can be determined from the measured photon count distribution without parity projection, we define the selective fidelity for single-atom ($N = 1$) detection (see Appendix)

$$F_1'(\bar{N}) \approx 1 - P(01) - \frac{P(12)}{1 - L_1}, \tag{3}$$

which takes into account that errors can occur due to mis-identification with either $N = 0$ or $N = 2$ (events with $N > 2$ are neglected). Note that unlike $F_1$ which depends only on the performance of the imaging system, $F_1'$ depends on the mean atom number $\bar{N}$.

Lastly we consider the selective fidelity for two-atom detection $F_2'$. Loss errors were calculated directly from the data. However calculation of the fidelity was hampered by the small number of frames with $N > 2$. Therefore the same threshold-based method was applied to the model instead, where the infidelity error can be obtained directly from the fraction of occurrences within thresholds B and C that were due to the model starting with two atoms. We checked that this model-based approach yielded values for $F_1'$ that were almost identical with those obtained empirically.

The resulting values of limiting infidelity and loss are shown in Fig. 5(b) and (c) as a function of cumulative exposure time, $t_{exp}$. In each case the thresholds are adjusted to optimise the fidelity. The uncertainty in the infidelity and loss is estimated by varying the thresholds by $\pm 1$ around the optimal value. Numerical values for an imaging time of 30 ms are provided in Table 1. A basic limit to the fidelity and loss that can be achieved is provided by depumping via the $5s5p\,^1P_1 \rightarrow 5s4d\,^1D_2$ transition. The corresponding loss probability was calculated by combining the branching ratio obtained in [15, 16] with an estimate of the scattering rate obtained from the measured intensity of the imaging beam. The resulting curves are in excellent agreement with the experimental data in Fig. 5(c). Together with the agreement with the Poisson model in Fig. 5(a), these data confirm that two-body loss due to light-assisted collisions is not significant in our experiment.

We also analysed the data as a time series for individual runs of the experiment. By applying the optimized thresholds used in Fig. 5(b) to each 10 ms frame, we obtain the variation of the trapped atom number in each experimental run. Examples of the results are shown in Fig. 5(d). One-body loss due to depumping is clearly visible as discrete downward steps in the atom number. As expected, events where both atoms are lost in the same frame are also observed. Infidelity events can be seen as upwards jumps in the atom number. Since there is no reservoir present to reload the tweezer, any events where the atom number increases must have had an incorrectly identified atom number either before or after the upward jump.

**Fast pulsed imaging without cooling**

The limiting fidelity that we achieve (for one atom) using the combined cooling and imaging method is comparable to that obtained in similar experiments where the 5s4d state is untrapped [15, 16] and repumping [21] is not possible. However close inspection of the data in Fig. 5 shows that for long exposures the dark count of the SPAD camera also plays a role. To investigate this effect, we explored imaging in the regime of a strong probe beam, such that sufficient photons are detected in a very short exposure.

Results for such an imaging technique, with an increased probe power ($S = 0.14$) are shown in Fig. 6. The measured histogram is in reasonable agreement with the composite Poisson model; here the mean atom number $\bar{N} = 0.5$ was lower than in Fig. 5 and the trap predominately contained either zero or one atom. The small number of two atom events precluded a reliable analysis of the two-atom loss and infidelity. The one-atom infidelity and loss are shown in Fig. 6(b) as a function of exposure time. Numerical values for the fidelity and loss with $t_{exp} = 200\,\mu s$ are provided in Table 1. The predicted loss to the $^1D_2$ state (solid line) shows that this imaging method, in the absence of any Sisyphus cooling, does not allow for this fundamental limit to be reached. Instead, the apparent loss is due to probe-induced heating, which drives the atom out of resonance with the probe beam leading to a reduction in the collected fluorescence signal at longer imaging times. This is highlighted by the inset shown in Fig. 6(b), which shows the counts on the SPAD as a function of imaging time. A simple heating model (dashed line) agrees with the experimental data, where it is assumed that scattering photons from the probe pushes the atom up the trap potential, reducing the differential AC Stark shift and shifting the atom out of resonance with the probe beam. However, even though significant heating takes place, this imaging method achieves a similar limiting fidelity to that obtained with Sisyphus cooling in Fig. 5, with a $t_{exp}$ that is a factor of 150 times shorter. This is due to the fact that the signal peak can be clearly resolved from the largely zero background obtained for empty traps (the mean number of background counts is 0.02 at $200\,\mu s$).

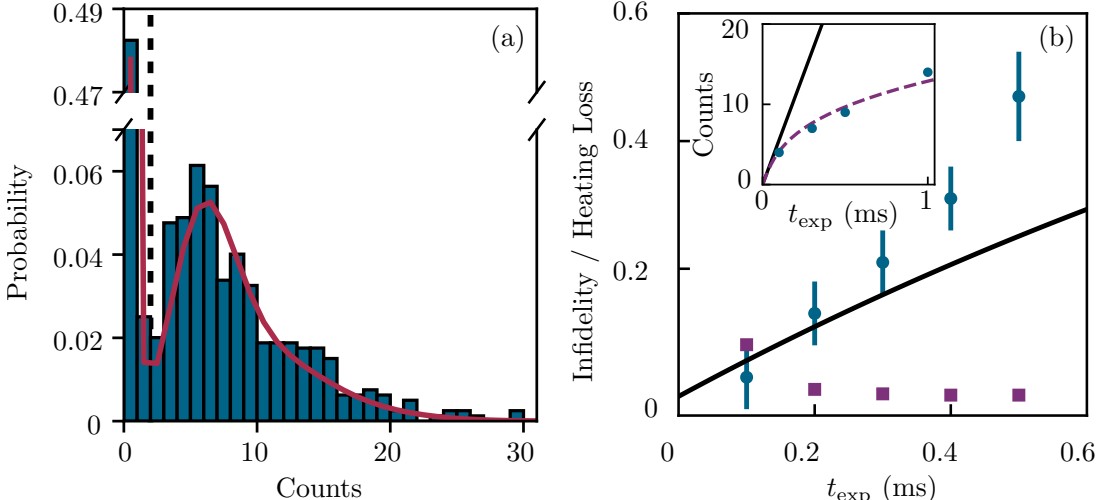

Figure 6: (a) Histogram of the number of counts for $t_{\mathrm{exp}} = 200$ μs and 800 experimental repetitions. The red solid line shows the composite Poisson distribution fit, and a black dashed line indicates the optimised threshold used to determine the infidelity and loss. (b) Limiting infidelity (purple squares) and heating induced loss (blue circles) as a function of imaging duration. The expected loss due to decay into the anti-trapped $^1\mathrm{D}_2$ state is also shown (black line). The inset shows the detected counts $n_c$ as a function of $t_{\mathrm{exp}}$ (blue circles). The solid line shows the corresponding prediction with (purple dashed line) and without (solid black line) heating.

**Pulsed imaging with Sisyphus cooling**

We therefore investigated whether it is possible to combine the advantages of both approaches, by applying a sequence of short, intense imaging pulses separated by periods of Sisyphus cooling. This pulsed imaging sequence is similar to the one introduced in [16], however here the fast readout speed of the camera is exploited by only analysing the frames that coincide with the imaging pulses. Therefore a much lower dark count can be obtained for a given number of detected signal photons. The results of such an experiment are shown in Fig. 7. Here the imaging beam (S = 0.14) was pulsed on for 41.6 μs with a repetition time of 4.16 ms. The exposure time for each frame was slightly longer (52 μs), and the camera frames and imaging pulses were carefully synchronized. The 689 nm cooling light was applied continuously during the imaging sequence at a detuning of +4.9 MHz. The combined histogram resulting from a total of 20 pulses or 832 μs of imaging time is shown in Fig. 7(a). The background and single-atom peaks are significantly better resolved than in either Fig. 5(a) or Fig. 6(a). With an optimal choice of threshold, the limiting (selective) fidelity is also improved reaching $F_1 = 0.998(2)$ ($F_1' = 0.991(2)$) with losses of $L_1 = 0.139(10)$ (Fig. 7(b)), at the expense of a longer total measurement time (imaging and cooling) of $\sim 83$ ms.

However, the losses shown in Fig. 7(b) cannot be explained solely by the expected D state loss (solid line). Both the loss, and the total number of detected photons are approximately a factor of three times lower than expected. This indicates that although cooling between the pulses has provided a significant advantage, it has not entirely compensated for the heating during the imaging pulses. Therefore it is likely that the loss and fidelity may be further improved with a more exhaustive study of the available parameter space. Since the optimum values are likely to depend on the differential AC Stark shift, and hence on the trap depth and wavelength, we leave this for future work.

Table. 1 summarises the different imaging techniques introduced in this section, comparing

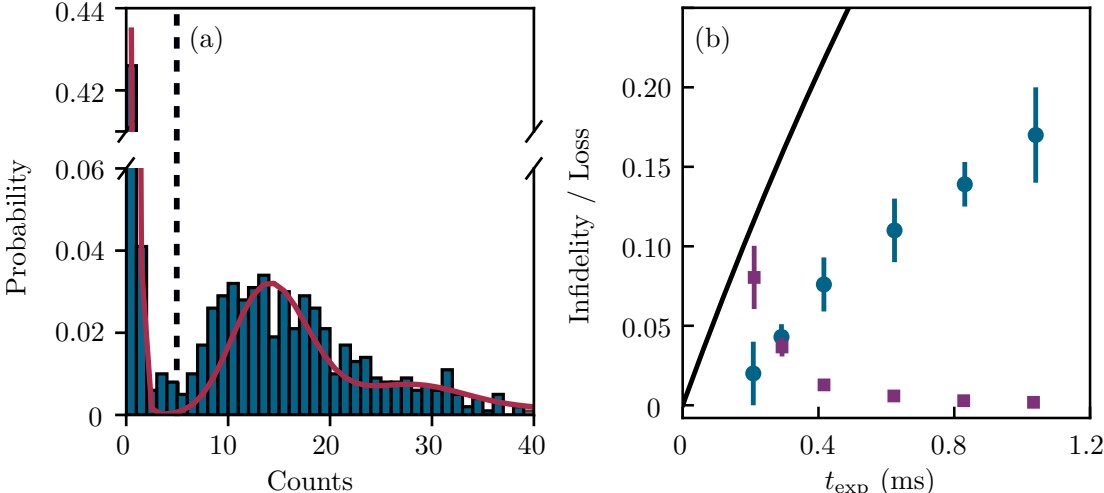

Figure 7: (a) Histogram of the number of counts after a $t_{\mathrm{exp}} = 832$ µs for 1000 experimental repeats. The red solid line shows the composite Poisson distribution model. The threshold used to determine the infidelity and loss is indicated by a dashed black line. (b) Limiting infidelity (purple squares) and losses (blue circles) as a function of $t_{\mathrm{exp}}$.

Table 1: Comparison of different imaging methods.

| Method | $t_{\mathrm{exp}}$ (ms) | $\bar{N}$ | $F_1$ | $F_1'(\bar{N})$ | $L_1$ | $F_2'(\bar{N})$ | $L_2$ |
|--------|-------------------------|-----------|-------|-----------------|-------|-----------------|-------|
| Fig. 5 | 30 | 1.2 | 0.989(3) | 0.970(2) | 0.211(8) | 0.83(2) | 0.373(11) |
| Fig. 6 | 0.2 | 0.5 | 0.989(6) | 0.979(6) | 0.13(5) | - | - |
| Fig. 7 | 0.83 (83) | 0.6 | 0.998(2) | 0.991(2) | 0.139(10) | - | - |

the fidelities ($F_1, F_{1,2}'$) and loss rates ($L_{1,2}$). The chosen $t_{\mathrm{exp}}$ is the same as for the histograms in Fig. 5 - 7, and was chosen as similar loss rates are observed at these times. For the pulsed method, the total time required for imaging (including the cooling) is stated in parenthesis. Two atom results are only included for continuous cooling and imaging (Fig. 5), as the mean atom number is too low for the other cases.

# 7  Discussion and Outlook

In conclusion, we have demonstrated that it is possible to trap and detect individual strontium atoms, with a fidelity of up to 0.998(2), in an optical tweezer with a working distance of 37 mm. In addition, we have generalised techniques for measuring the temperature and trap parameters developed for alkali-metal atoms. Surprisingly, we find that loading from a MOT working on a narrow intercombination line is very effective, enabling cold samples of atoms to be prepared in trap depths of up to 7.5 mK. An important element of our experiment is the use of a camera based on an array of single-photon counting detectors. We found that the low readout noise and high frame rate of this detector enabled high fidelity detection of single atoms without cooling at very short timescales (200 µs vs 30 ms).

The performance of our experiment was also sufficient for high-fidelity time and number resolved detection of $N = 0, 1, 2$ atoms in our tweezer. Monitoring the atom number in real time could potentially provide a way to deterministically prepare a single atom. To do so, several challenges must be overcome. Firstly, the probability $P(0)$ of loading no atoms must

be minimised. For $\bar{N} = 1.2$ as in Fig. 5, $P(0) = 0.3$. This method could therefore potentially be competitive with the state-of-the-art demonstrated for Sr of $P(0) = 0.5$ [16, 21]. Reaching the current state-of-the-art for alkali-metal atoms (without transport) of $P(0) = 0.1$ would require $\bar{N} = 2.3$. Secondly, the fidelity and loss must be further improved to reduce the errors and the probability of losing more than one atom in each frame. Switching to 813 nm where the the $^1D_2$ state is trapped and repumping is possible would enable much higher fidelity [21], as well as control over the one-body loss via the repumping process. Therefore it appears feasible that time-resolved measurements of the trapped atom number could provide a useful tool to enhance the loading probability in future experiments.

Overall, these results clearly demonstrate that it is possible to construct and operate an optical tweezer experiment with a much lower numerical aperture and a much longer working distance than that employed in previous experiments. In future experiments, creating a tweezer array and combining the methods shown here with techniques such as Rydberg dressing [45, 46], could form an ideal platform for testing proposals to create highly entangled states of strontium atoms [25, 47], as well as other avenues such as precision measurement of Rydberg states [48] and transport effects [49].

## Acknowledgements

We thank M. Endres and A. Browaeys for technical information. We also acknowledge the contribution of Dr P. Huillery to the early stages of the project.

**Author contributions**     N. C. Jackson and R. K. Hanley contributed equally to this work.

**Funding information**     This work was supported by EPSRC Responsive Mode grants EP/R035482/1 and EP/J007021/1. NCJ was supported by EPSRC Platform grant EP/R002061/1. The project acknowledges funding of the project EMPIR-USOQS, EMPIR projects are co-funded by the European Union's Horizon2020 research and innovation programme (640378-RYSQ) and the EMPIR Participating States (17FUN03-USOQS).

## Appendix

Here we generalise the concept of a single-atom infidelity error to the case where more than one atom may be present. Let an error occur if a single atom ($N = 1$) is identified in the second frame but not the first, with associated probability

$$P(\bar{1}1) = P(01) + P(21) + P(m1), \tag{4}$$

where $\bar{1}$ means $N \neq 1$, and $m$ refers to all events with $N > 2$ for which we do not attempt to resolve atom number.

In previous work, errors due to loss are treated separately from infidelity errors. To separate out the contribution of loss to the $P(21)$ term we replace $P(21)$ with $P(12)$, as the two values should be similar in the absence of loss. Directly swapping these values underestimates the number of infidelity events, as the probability of seeing events with one atom in the first frame and two in the second are suppressed by loss. Therefore P(12) is rescaled by the loss rate to prevent undercounting. Neglecting the $P(m1)$ terms, where the atom number changes by 2 or more between the first and second frame, gives an expression for the selective fidelity

of

$$F_1'(\bar{N}) \approx 1 - P(01) - \frac{P(12)}{1 - L_1}.$$ (5)

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
