# Peer review of "Number-resolved imaging of $^{88}$Sr atoms in a long working distance optical tweezer"

_SciPost Physics, doi:SciPost Phys. 8, 038 (2020)_

## Round 2 · Referee Report · Anonymous (Referee 1) · 2019-5-8

Strengths

See attached document below.

Weaknesses

See attached document below.

Report

See attached document below.

Requested changes

See attached document below.

Attachment

  • validity: ok
  • significance: low
  • originality: ok
  • clarity: ok
  • formatting: reasonable
  • grammar: good

Author:  Matthew Hill  on 2019-10-02  [id 617]

(in reply to Report 1 on 2019-05-08)

Please see attached file

Attachment:

Tweezer_Response.pdf

---

## Round 2 · Referee Report · Anonymous (Referee 2) · 2019-5-10

Strengths

See report

Weaknesses

See report

Report

The authors present initial studies of imaging and temperature for strontium atoms confined within an optical tweezer formed with a relatively low NA lens. The key claims to novelty are:

1) The long working-distance of the system is a key attribute for future studies involving Rydberg states 2) Number resolved imaging can be achieved 3) Parity projection can be avoided in the larger-volume trap. 4) The use of a SPAD array detector for atomic detection, providing performance advantages over typical CCD’s.

In my opinion, these claims all require further quantitative justification. In order: 1) This would be a nice feature to have, if NA were not also severely sacrificed. Currently, there are major demonstrated advantages to a high-NA setup (reduced optical power requirements, lower effects of atomic motion, possibility of sideband cooling, higher collection efficiency), but no demonstration that setups with ~5mm separation between atoms and glass surfaces are incompatible with Rydberg physics.

2) No quantitative demonstration of number resolved imaging is presented. This claim cannot be made without providing the fidelity with which different number states can be distinguished, and the measured rate of atom loss associated with imaging at this fidelity. Further, it is claimed that this imaging technique can be used to to deterministically prepare loaded tweezers. In estimating the fidelity of this approach, no mention is made of the imaging fidelity, which would limit one’s ability to load single atoms.

3) No quantitative measurement of parity projection avoidance is presented. The authors claim that parity projection is avoided by using a larger trap, but no quantitative information is presented to substantiate this claim. Further, the authors argue that the lack of parity projection in their system contrasts with previous strontium tweezer experiments. However, as indicated in the appendices of references 15, 16, substantially longer parity projection steps are used to isolate single atoms, presumably because parity projection actually takes a while in those systems as well.

4) The authors claim that the use of a SPAD array provides advantages over CCD or CMOS cameras, and specifically that unlike CCD or CMOS cameras, SPAD arrays are limited by photon shot noise instead of dark counts or readout noise. However, the cooled EMCCD cameras used in tweezer and quantum gas microscope experiments are typically dominated by shot noise (with a contribution from the excess noise factor associated with gain, though this can also be avoided), and have much lower dark count rates than those of the SPAD array, much higher quantum efficiency, and negligible readout noise. There may be a cost advantage to the SPAD array, but if this is the reason to use them then it should be stated.

Other issues: • figure 3a: I think the X axis here is supposed to be microseconds, not milliseconds. After 50ms, gravity would have caused the atoms to drop by over 1 cm.
• figure 3b: Perhaps there is an issue with the axes here as well. The line implies a velocity of around 50m/s, which is not what one would expect for the quoted temperatures.
• Are the temperatures quoted in the axial or radial directions, or are the two assumed to be equal?

Requested changes

see report

  • validity: ok
  • significance: ok
  • originality: ok
  • clarity: ok
  • formatting: good
  • grammar: excellent

Author:  Matthew Hill  on 2019-10-02  [id 618]

(in reply to Report 2 on 2019-05-10)

Please see attached file

Attachment:

Tweezer_Response_9ETBMu2.pdf

---

## Round 3 · Referee Report · Anonymous (Referee 4) · 2019-10-8

Strengths

See report.

Weaknesses

See report.

Report

This manuscript has been significantly improved, and I believe it now makes a very valuable addition to the growing community of atom-resolved research with alkaline-earth (-like) atoms. The authors addressed all of my concerns in great detail, and I think their narrative is much more clear and substantiated in the new version. I would now recommend publication. However, please find a few minor change requests below.

Requested changes

1. I would caution against the final sentence of the abstract, which says "these results show that high NA lenses are not an essential requirement for optical tweezer experiments." Certainly this statement needs further qualification. At face value, I think it is widely known that high NA is not strictly necessary for single-atom control and detection. However, for most Rydberg-based physics goals - particularly outside of the blockade regime - a high degree of atomic localization is essential, and this requires tight optical traps generated with high NA objectives. For assembled Hubbard systems, the requirement on high NA is thought to be even higher since tunneling must be controlled. My point is only that it depends on the goals of the experiment.
2. In the first sentence of the third paragraph in Section 4, the authors say "very deep tweezers". Please add a number here, even if it is rough. The authors use depths in uK units later in that paragraph, so I would like a similar estimate to quantify "very deep".
3. The discussion in that same paragraph on subsidiary intensity maxima is a bit surprising to me. Can you quantify the relative depth of these additional maxima? Is that estimate consistent with the alleged Airy disk pattern induced by the circular aperture of the lens. What about the size profile? The authors suggest they have some spatial resolution of this effect. More quantitative detail would be helpful here since this is perhaps a new observation for Sr.
4. A few paragraph later, starting with "A typical release and recapture signal...": what is the trap depth corresponding to that temperature measurement and estimate? Would it be appropriate to estimate radial n (motional quanta)?
5. In Figure 4, what is 'N' as the vertical axis of the inset plots? Presumably This is related to atom survival, as described in the text. Perhaps the axis should be normalized to unity?
6. In Figure 5a, I would suggest adding the relative contribution of the four sections. Presumably this is consistent with the Nbar=1.2 stated in table 1.
7. In the discussion of pulsed detection where blue scattering is pulsed and red Sisyphus cooling is always on: My understanding of the initial observations of this "repulsive" Sisyphus mechanism is that it can repel the atom both below and above a critical energy. Is it possible that the blue scattering quickly heats the atom above the critical energy with some probability, after which it is lost since the "cooling" only hurts in that case? Further, if the atom is not near the critical energy, the "cooling" largely does nothing. Is it really necessary to use the "cooling" during the time when the blue pulse is off?
8. Finally, I believe the reader would appreciate a stronger and more clear final outlook. What exactly are the intended goals of this experiment, and what further directions would benefit most from its unique features? The authors mentioned precision optical metrology in their response letter, but no such statement is explicitly made in the text, although it is perhaps implied in the last sentence.

  • validity: high
  • significance: good
  • originality: high
  • clarity: high
  • formatting: excellent
  • grammar: excellent

Author:  Matthew Hill  on 2020-01-13  [id 704]

(in reply to Report 1 on 2019-10-08)

Please see attached file.

Attachment:

Referee_1_Response.pdf

---

## Round 3 · Referee Report · Anonymous (Referee 3) · 2019-10-22

Strengths

see report

Weaknesses

see report

Report

The authors have significantly improved this manuscript since the first submission.  In particular, the core claims are now supported in a quantitative manner, and  the relative merits of the SPAD array versus other imaging devices is now clearly discussed and demonstrated.  

I have one remaining significant concern with the manuscript, which is the definition of fidelity used, and the fidelity numbers presented using these definitions.  The authors adopt methods for calculating infidelities from references 15, 16, 21, in which the infidelity is inferred from the probability of observing an void in the first of two images and an atom in the second.  This method for inferring the infidelity is not general, and rests upon assumptions that are not valid here, such as the presence of only either zero or one atom in the tweezer (see for example appendix B5 of ref 16, in particular eq. B1).  
As an example of the problems that are caused by ignoring this assumption, in the submitted manuscript, P0->1 is used as a proxy for the single atom infidelity, but does not incorporate the probability that two atoms initially occupying the tweezer are mistakenly identified as one.  Visual inspection of the red curve in figure 5a indicates that the contributions from one, two, and three atoms are significantly overlapped, so claiming infidelities at the 1-2% level seems highly dubious here.  Since the quantitative values claimed for fidelity are central to this work, I think that it is critical to provide a precise definition of what is meant by fidelity (which should include all ways in which a given atom number may be misidentified, such an initial double occupancy being mistaken for a single), and a clear derivation of how this definition is related to experimentally observed quantities.

Requested changes

see report

  • validity: ok
  • significance: ok
  • originality: good
  • clarity: high
  • formatting: excellent
  • grammar: excellent

Author:  Matthew Hill  on 2020-01-13  [id 705]

(in reply to Report 2 on 2019-10-22)

Please see attached file.

Attachment:

Referee_2_Response.pdf

---

## Round 3 · Author Response

Please see response to referees.

---

## Round 4 · Referee Report · Anonymous (Referee 1) · 2020-1-14

Report

I am satisfied with the authors' response to my previous comments, and I now recommend publication.

---

## Round 4 · Referee Report · Anonymous (Referee 2) · 2020-1-23

Report

The authors have substantially improved the calculation and discussion of imaging fidelity. My one remaining request would be to mention in the abstract the degree of loss associated with the quoted fidelity. Generally, loss and fidelity limit the performance of an experiment in a similar manner, so it is important to know both. In particular, past experiments have typically optimized for simultaneously low loss and high fidelity, so knowing both numbers is critical for comparison. If loss is not mentioned, the reader may assume that it has been included in the definition of fidelity. Apart from that small but I think important point, I now recommend the manuscript for publication.

Requested changes

see above

---

## Round 4 · Author Response

Please see responses to referees for changes.

---

## Round 5 · List of Changes

Value of loss added to abstract

---

## Editorial Decision

published